# Systematic Identification of CpxRA-Regulated Genes and Their Roles in *Escherichia coli* Stress Response

Zhe Zhao,[a,b] Ying Xu,[b,c] Bo Jiang,[a] Qingsheng Qi,[a] Ya-Jie Tang,[a] Mo Xian,[b] Jichao Wang,[a,b] Guang Zhao[a,b]

[a]State Key Laboratory of Microbial Technology, Shandong University, Qingdao, China
[b]CAS Key Laboratory of Biobased Materials, Qingdao Institute of Bioenergy and Bioprocess Technology, Chinese Academy of Sciences, Qingdao, China
[c]Navy Submarine Academy, Qingdao, China

**ABSTRACT** The two-component system CpxRA can sense environmental stresses and regulate transcription of a wide range of genes for the purpose of adaptation. Despite extensive research on this system, the identification of the CpxR regulon is not systematic or comprehensive. Herein, genome-wide screening was performed using a position-specific scoring matrix, resulting in the discovery of more than 10,000 putative CpxR binding sites, which provides an extensive and selective set of targets based on sequence. More than half of the candidate genes ultimately selected (73/97) were experimentally confirmed to be CpxR-regulated genes through experimental analysis. These genes are involved in various physiological functions, indicating that the CpxRA system regulates complex cellular processes. The study also found for the first time that the CpxR-regulated genes *ydeE*, *xylE*, *alx*, and *galP* contribute to *Escherichia coli* resistance to acid stress, whereas *prlF*, *alx*, *casA*, *yacH*, *ydeE*, *sbmA*, and *ampH* contribute to *E. coli* resistance to cationic antimicrobial peptide stress. Among these CpxR-regulated genes, *ydeE* and *alx* responded to both stressors. In a similar way, a cationic antimicrobial peptide is capable of directly activating the periplasmic domain of CpxA kinase *in vitro*, which is consistent with the CpxA response to acid stress. These results greatly expand our understanding of the CpxRA-dependent stress response network in *E. coli*.

**IMPORTANCE** CpxRA system is found in many pathogens and plays an essential role in sensing environmental signals and transducing information inside cells for adaptation. It usually regulates expression of specific genes in response to different environmental stresses and is important for bacterial pathogenesis. However, systematically identifying CpxRA-regulated genes and elucidating the regulative role of CpxRA in bacteria responding to environmental stress remains challenging. This study discovered more than 10,000 putative CpxR binding sites based on sequence. This bioinformatics approach, combined with experimental assays, allowed the identification of many previously unknown CpxR-regulated genes. Among the novel 73 CpxRA-regulated genes identified in this study, the role of nine of them in contributing to *E. coli* resistance to acid or cationic antimicrobial peptide stress was studied. The potential correlation between these two environmental stress responses provides insight into the CpxRA-dependent stress response network. This also improves our understanding of environment-bacterium interaction and Gram-negative pathogenesis.

**KEYWORDS** CpxRA two-component system, CpxR-regulated genes, position-specific scoring matrix, stress response, *Escherichia coli*

In bacteria, the two-component system (TCS) plays an essential role in sensing environmental stresses and transducing the information inside the cells for adaptation in bacteria (1). The TCS is basically composed of a sensor histidine kinase (inner membrane protein) and a cognate response regulator (cytoplasm protein) (1). CpxRA is a

Address correspondence to Guang Zhao, zhaoguang@sdu.edu.cn, or Jichao Wang, wangjichao@ustc.edu.

The authors declare no conflict of interest.

well-characterized TCS involved in envelope stress responses, consisting of CpxA (sensor histidine kinase) and CpxR (response regulator). In several Gram-negative bacteria, it contributes to environmental adaptation, e.g., intestinal infections (2, 3), heavy metal tolerance (4), virulence (5), antibiotic resistance (6), acid tolerance (7), biofilm formation (8), and oxidative stress (9). As in most TCSs, CpxA can phosphorylate or dephosphorylate its cytoplasmic cognate partner response regulator, CpxR (10). A variety of environmental signals can activate CpxA, resulting in its autophosphorylation using ATP at a conserved histidine residue. Subsequently, the phosphoryl group is transferred to an aspartate residue on CpxR. Finally, the phosphorylated CpxR (CpxR-P) regulates the expression of target genes involved in protection against environmental stress. Three genes were initially identified that were regulated by CpxR—*dsbA*, *degP*, and *ppiA* (11, 12)—followed by a growing number of other CpxR-regulated genes involved in different Cpx responses. The CpxRA TCS integrates physical, chemical, and biological signals, indicating its underlying role for biological processes in Gram-negative bacteria (2, 13–16).

The CpxRA TCS usually regulates specific gene expression in response to different signals. For example, in *Salmonella*, CpxRA can respond to gold (Au) ions and promotes *gesABC* transcription to protect cells from Au damage (4). In *Pseudomonas aeruginosa*, CpxRA response to antibiotic stress by activating *mexAB-oprM* expression, which is important for multidrug resistance (17). Moreover, we demonstrated previously that CpxRA directly senses acidification through protonation of CpxA periplasmic histidine residues and promotes *fabA* and *fabB* transcription to improve *Escherichia coli* survival under mild acid conditions (7). These results support the existence of a complex *E. coli* stress response network dependent on the CpxRA TCS. In addition to these individual TCSs responding to complex external environment changes, Oshima et al. (18) proposed the presence of functional interactions between different TCSs, such as cross talk and cascades of signal transductions. These would enable *E. coli* to fine-tune its environmental adaptability and favor the survival of bacteria (1).

To date, more than 40 CpxR-regulated genes have been found in bacteria. These target genes facilitate the discovery of CpxRA-mediated stress responses and are probably putative links between the CpxRA system and other signal transduction pathways. For example, the CpxR-regulated gene *acrD*, which encodes a multidrug efflux pump RND permease, is also regulated by other two-component systems, such as EvgAS and BaeSR (19, 20). In addition, CpxR appears to function to enhance the expression of *acrD* mediated by BaeR (20). These findings provide a foundation for further research into the CpxRA-mediated bacterial multidrug resistance mechanism, as well as evidence that two envelope stress response systems may work together to combat environmental stress by coregulating the expression of some target genes. As a result, identifying more CpxR-regulated genes will facilitate our understanding of the CpxRA two-component regulatory system in response to a wide variety of environmental stress and the complicated cross talk between the CpxRA system and other stress response pathways. However, screening and identifying more target genes, elucidating the regulative role of the CpxRA system, and illustrating the mechanisms by which functional interactions are established between CpxRA and other stress response systems represent a challenging endeavor.

In this study, we screened the *E. coli* BW25113 genome sequence based on the position-specific scoring matrix (PSSM) and discovered more than 10,000 putative CpxR binding sites in promoter regions which are similar to the CpxR motif (GTAAA-N$_{5-6}$-GTAAA) (21). Of these, 97 candidate genes were selected for transcriptional analysis. To our knowledge, none of these genes have been reported to be regulated by CpxR. More than half of these genes (73/97) were identified to be regulated by the CpxRA system, at least under one condition of CpxRA activation. This is more than the overall number available accumulated in the past decades. This discovery increases our understanding of the overall Cpx pathway-dependent environmental stress response. These CpxR-regulated genes are functionally diverse and involved in complex physiological processes.

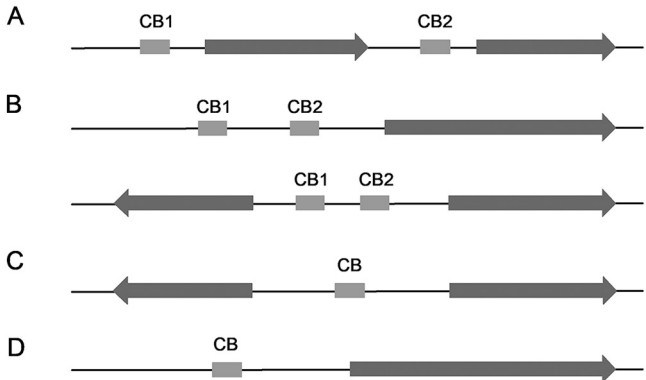

**FIG 1** Schematic diagram illustrating the four groups of putative CpxR-regulated genes obtained by PSSM. (A) Multiple putative CpxR boxes exist within gene clusters. (B) Multiple putative CpxR boxes exist at promoter regions of the gene. (C) One putative CpxR box exists at intergenic regions. (D) One putative CpxR box exists at promoter regions of the gene. Predicted CpxR binding sequence (gray box) and candidate gene (brown area) are shown.

Gel shift analyses revealed that some of them are controlled by CpxR direct binding to the promoters. The contribution of nine of these 73 target genes to *E. coli* resistance to either acid or protamine (a model cationic antimicrobial peptide) was investigated. We determined that *ydeE* and *alx* participate in both environmental stress responses. Furthermore, analysis of the reconstituted proteoliposome revealed that the periplasmic domain of CpxA kinase acts like a sensor domain of protamine and acid. These results further support the idea that CpxRA connects different environmental stress responses by varying the expression of specific target genes, which is responsible for mobilizing subsequent programs and thus improving bacteria's adaptability to environmental stress.

## RESULTS

**Screening the possible CpxR binding sites using PSSM.** In this study, we attempted to determine the contribution of the individual bases by PSSM (22, 23) using the previously reported CpxR recognition sequence GTAAA-$N_{5-6}$-GTAAA as a reference (21). This can help guide future research into identifying putative CpxR-regulated genes. Based on 41 known CpxR binding sites (see Table S1 in the supplemental material), the PSSM sources of 15-bp sequences were produced. To compensate for the lack of data for 16-bp CpxR binding sites, PSSM sources of 16-bp sequences were obtained by repeating the middle base of known 15-bp CpxR binding sites. In this study, the *E. coli* BW25113 genome was scanned for potential CpxR boxes, focusing on the promoter and adjacent regions of putative CpxR-regulated genes (700 bp upstream to 100 bp downstream of the start codons). We hypothesized that sequences with higher PSSM scores have higher information entropy and would be more likely to interact with CpxR protein physically. In total, 6,522 conserved 15-bp and 6,464 conserved 16-bp sequences were found (threshold = mean of PSSM output scores − standard deviation of PSSM output scores) (Tables S2 and S3). These candidate genes were classified into four groups based on the location of their potential CpxR box (Fig. 1). From these four groups, 97 candidate genes from four groups were randomly chosen (Table S4), and their promoter regions contained conserved sequences similar to a CpxR recognition site. To the best of our knowledge, none of these candidate genes have been shown to be regulated by CpxR before.

**Identification of CpxR-regulated genes.** The outer membrane-anchored lipoprotein NlpE acts as an activator of the CpxRA system when overexpressed (24). Furthermore, the *cpxA24* allele (which has a deletion which encompassing 32 amino acids in the central region of the periplasmic loop) also results in the constitutive activation of the Cpx response independently of any inducing cues (25). Both of these

**TABLE 1** Transcriptional analysis of known CpxR-regulated genes

| Gene | Function | Proposed *cpx* regulation | Avg fold difference in expression | | | |
|------|----------|---------------------------|----------|--------|-------|--------|
| | | | cpxA24 | ΔcpxA | pnlpE | ΔcpxR pnlpE |
| *degP* | Periplasmic serine endoprotease | Positive (12) | 26.60 | 2.91 | 32.41 | 6.43 |
| *htpX* | Protease | Positive (53) | 25.23 | 17.94 | 37.59 | 0.61 |
| *cpxP* | Periplasmic adaptor protein | Positive (54) | 18.99 | 7.59 | 60.27 | 0.03 |
| *ftnB* | Putative ferritin-like protein | Positive (55) | 2.05 | 1.39 | 3.24 | 0.49 |
| *sbmA* | Peptide antibiotic/peptide nucleic acid transporter | Positive (15) | 5.75 | 0.94 | 3.20 | 1.69 |
| *srkA* | Stress response kinase A | Positive (56) | 3.49 | 1.40 | 0.90 | 1.81 |
| *tsr* | Methyl-accepting chemotaxis protein | Negative (26, 27) | 5.01 | 2.06 | 0.79 | 1.09 |
| *yccA* | Modulator of FtsH protease | Positive (55) | 3.96 | 3.62 | 4.89 | 0.03 |
| *slt* | Soluble lytic murein transglycosylase | Positive (15) | 6.57 | 3.32 | 5.07 | 0.31 |
| *alx* | Putative membrane-bound redox modulator | Positive (21) | 5.36 | 3.55 | 4.61 | 2.06 |
| *ompC* | Outer membrane porin C | Positive (57) or no difference (15, 27) | 1.23 | 0.82 | 1.16 | 0.02 |
| *efeU* | Inactive ferrous iron permease | Negative (27, 58) | 0.53 | 0.52 | 0.02 | 2.61 |
| *amiC* | N-Acetylmuramoyl-L-alanine amidase C | Positive (43) | 1.53 | 0.83 | 0.87 | 0.84 |
| *psd* | Phosphatidylserine decarboxylase | Positive (21) | 8.46 | 2.58 | 2.03 | 0.71 |
| *motA* | Flagellar motor component | Negative (21) | —[a] | — | 0.44 | 1.38 |

[a]— means insignificant differences (< 2-fold) in the transcriptional levels.

circumstances are frequently used to activate the Cpx pathway. We used quantitative reverse transcription-PCR (qRT-PCR) to examine the expression during the log phase of several previously reported CpxR-regulated genes that were placed in two groups according to genetic background. One group was formed by BW25113, the *cpxA24* strain, and a Δ*cpxA* mutant. The other group had BW25113 with the empty vector and recombinant plasmid carrying *nlpE* and a Δ*cpxR* mutant with a recombinant plasmid carrying *nlpE*. In general, the transcriptional regulation of these genes is consistent with previous reports, under one or both of the activating conditions (Table 1). However, with one exception, although the weakly negative regulation by the Cpx response was detected after NlpE overexpression in this study, *tsr* was positively regulated in the *cpxA24* background, which is in contrast with previous studies (26, 27). These results suggest that these two activation conditions can be used to identify CpxR-regulated genes, although expression levels or expression pattern may not be completely consistent under different activation conditions.

Then, the expression of the 97 candidate genes obtained was measured in two groups separated by genetic background. We observed significant differences (≥2-fold) in the transcriptional levels of 42 genes in the *cpxA24* mutant and the Δ*cpxA* mutant compared to the wild type (Table 2). These genes are involved in a wide range of physiological functions, such as amino acid synthesis and degradation, electron transfer, $H^+$ transport, central metabolism, iron acquisition, quorum sensing, biofilm function, and stress responses. The majority of proteins encoded by these genes are located in the cytoplasm and inner membrane. Interestingly, in the Δ*cpxA* mutant, some Cpx-regulated genes still showed a slight response, which could be due to the leaky output caused by the loss of CpxA phosphatase activity in the Δ*cpxA* mutant and the phosphotransfer from acetyl phosphate (acetyl-P) to CpxR. Indeed, this intermediate of the phosphotransacetylase (Pta)-acetate kinase (AckA) pathway can donate its phosphoryl group to CpxR without CpxA (28).

To test this hypothesis, we measured the mRNA levels of some CpxR-regulated genes in wild-type (WT) cells and isogenic mutants, including (i) a Δ*cpxA* mutant, (ii) a Δ*cpxR* mutant, (iii) a Δ*cpxA* Δ*cpxR* mutant, (iv) a Δ*pta* Δ*ackA* mutant, (v) a Δ*cpxA* Δ*pta* Δ*ackA* mutant, and (vi) a *cpxA24* mutant. In this study, the Δ*cpxA* mutant exhibited leaky output compared with the wild type. Intriguingly, the Δ*cpxA* Δ*pta* Δ*ackA* triple mutant exhibited a dramatic decrease in this leaky output compared with the Δ*cpxA* mutant (Fig. 2). These results indicated that disruption of the Pta-AckA pathway diminished Cpx signaling and support the hypothesis that CpxR-P would accumulate and cause a leaky output because of the donation of a phosphoryl group from the Pta-AckA pathway coupled with the deletion of phosphatase activities of CpxA. As expected, deleting *cpxR* or removing the CpxRA system completely eliminated this leaky output (Fig. 2). However, for *htpX* and

**TABLE 2** Transcriptional analysis of identified CpxR-regulated genes

| Gene category | Gene | Function[a] | Cellular location | Avg fold difference in expression | | | |
| --- | --- | --- | --- | --- | --- | --- | --- |
| | | | | cpxA24 | ΔcpxA | pnlpE | ΔcpxR pnlpE |
| Amino acid transport and metabolism | carA | Carbamoyl phosphate synthetase subunit alpha | Cytoplasmic | 42.76 | 1.16 | 5.70 | 0.42 |
| | carB | Carbamoyl phosphate synthetase subunit beta | Cytoplasmic | 31.80 | 3.02 | 34.10 | 5.20 |
| | glsA | Glutaminase 1 | Cytoplasmic | 0.08 | 0.29 | 0.01 | 0.13 |
| | metC | Cystathionine beta-lyase/L-cysteine desulfhydrase | Cytoplasmic | 0.40 | 0.22 | 0.08 | 0.06 |
| | edd | Phosphogluconate dehydratase | Cytoplasmic | 2.26 | 0.61 | 0.77 | 0.44 |
| | astC | Succinylornithine transaminase | Cytoplasmic | —[b] | — | 0.13 | 0.25 |
| | epmB | Lysine 2,3-aminomutase | Cytoplasmic | 2.43 | 1.78 | 0.48 | 0.53 |
| | yhdW | Putative ABC transporter periplasmic binding protein | Periplasmic | 0.12 | 0.25 | 0.29 | 2.18 |
| | sdaC | Amino acid permeases | Inner membrane | — | — | 0.31 | 0.28 |
| Energy production and conversion | fdnG | Formate dehydrogenase N subunit alpha | Periplasmic | 7.34 | 1.41 | 0.59 | 3.01 |
| | hcp | Protein S-nitrosylase | Cytoplasmic | 2.37 | 1.70 | 0.15 | 1.03 |
| | appC | Cytochrome bd-II ubiquinol oxidase subunit I | Cytoplasmic | 0.18 | 0.17 | 0.16 | 0.28 |
| | atpI | ATP synthase accessory factor | Inner membrane | 3.42 | 1.25 | 2.88 | 1.19 |
| | cyoA | Cytochrome o ubiquinol oxidase subunit II | Inner membrane | — | — | 0.32 | 0.29 |
| | frc | Formyl-CoA transferase | Cytoplasmic | — | — | 6.18 | 1.26 |
| | atpB | $F_oF_1$-type ATP synthase, subunit a | Inner membrane | 4.95 | 3.51 | 0.51 | 0.3 |
| Inorganic ion transport and metabolism | nrfA | Cytochrome c552 nitrite reductase | Periplasmic | 10.25 | 3.57 | — | — |
| | chaA | $Na^+/K^+$:$H^+$ antiporter | Inner membrane | 44.89 | 13.10 | 5.37 | 0.57 |
| | chaB | Putative cation transport regulator | Cytoplasmic | 0.21 | 0.22 | 0.26 | 1.90 |
| | copA | Soluble $Cu^+$ chaperone | Inner membrane | 2.06 | 1.21 | — | — |
| | fetA | Putative iron ABC exporter ATP-binding subunit | Inner membrane | 0.23 | 0.26 | 0.30 | 0.29 |
| | xylE | D-Xylose:$H^+$ symporter | Inner membrane | — | — | 0.80 | 3.59 |
| | focA | Formate channel | Inner membrane | — | — | 0.72 | 0.06 |
| | ybaL | Thiamine transporter subunit | Periplasmic | — | — | 0.25 | 1.04 |
| Carbohydrate transport and metabolism | gudP | Galactarate/glucarate/glycerate transporter | Inner membrane | 0.42 | 1.91 | — | — |
| | ydeE | Dipeptide exporter | Inner membrane | 0.38 | 0.50 | 0.28 | 1.36 |
| | araF | Arabinose ABC transporter periplasmic binding protein | Periplasmic | 1.13 | 0.27 | — | — |
| | yddG | Aromatic amino acid exporter | Inner membrane | 2.39 | 1.38 | 0.42 | 0.98 |
| | eamA | Cysteine/O-acetylserine exporter | Inner membrane | 0.44 | 0.58 | 0.27 | 0.78 |
| | gpmM | 2,3-Bisphosphoglycerate-independent phosphoglycerate mutase | Cytoplasmic | 4.92 | 0.82 | 0.15 | 0.03 |
| | frmB | S-Formylglutathione hydrolase | Cytoplasmic | — | — | 0.46 | 0.75 |
| | gapC | Glyceraldehyde-3-phosphate dehydrogenase (pseudogene) | Unknown | — | — | 0.17 | 0.06 |
| | gmhA | D-Sedoheptulose 7-phosphate isomerase | Cytoplasmic | — | — | 0.26 | 0.46 |
| | galP | Galactose:$H^+$ symporter | Inner membrane | — | — | 0.58 | 3.58 |
| | ascF | Beta-glucoside-specific PTS enzyme IIBC component | Inner membrane | — | — | 0.47 | 0.84 |
| Signal transduction mechanisms | btsT | Pyruvate:$H^+$ symporter | Inner membrane | — | — | 0.06 | 1.01 |
| | ylaB | Predicted cyclic-di-GMP phosphodiesterase | Cytoplasmic, Inner membrane, Periplasmic | 0.42 | 0.53 | 0.37 | 0.98 |

**TABLE 2** (Continued)

| Gene category | Gene | Function[a] | Cellular location | Avg fold difference in expression | | | |
|---|---|---|---|---|---|---|---|
| | | | | cpxA24 | ΔcpxA | pnlpE | ΔcpxR pnlpE |
| Transcription | bluR | DNA-binding transcriptional repressor | Cytoplasmic | 0.25 | 0.40 | 0.25 | 0.26 |
| | fis | DNA-binding transcriptional dual regulator | Cytoplasmic | 3.98 | 2.07 | — | — |
| | ettA | Energy-dependent translational throttle protein | Cytoplasmic | 2.43 | 1.76 | 0.50 | 0.28 |
| | cspA | Cold shock protein | Cytoplasmic | — | — | 0.86 | 0.33 |
| | prlF | Antitoxin | Cytoplasmic | — | — | 0.25 | 0.33 |
| | cbl | DNA-binding transcriptional activator | Cytoplasmic | — | — | 11.85 | 98.39 |
| | ecpR | DNA-binding transcriptional dual regulator | Cytoplasmic | — | — | 0.90 | 6.44 |
| | feaR | DNA-binding transcriptional activator | Cytoplasmic | — | — | 0.36 | 1.14 |
| Intracellular trafficking, secretion, and vesicular transport | exbB | Ton complex subunit | Inner membrane | — | — | 0.70 | 0.20 |
| Defense mechanisms | ampH | Peptidoglycan DD-carboxypeptidase/peptidoglycan DD-endopeptidase | Periplasmic | 3.45 | 2.58 | 2.21 | 0.72 |
| | casA | Type I-E CRISPR system Cascade subunit | Cytoplasmic | 4.16 | 1.06 | 3.76 | 0.83 |
| | shoB | Toxic peptide | Inner membrane | 0.37 | 0.60 | 0.09 | 3.28 |
| | inaA | Putative lipopolysaccharide kinase | Cytoplasmic | — | — | 3.38 | 1.26 |
| Posttranslational modification, protein turnover, chaperones | qmcA | PHB domain-containing protein | Inner membrane | 0.21 | 0.43 | 0.21 | 1.26 |
| Cell wall/membrane/envelope biogenesis | dgkA | Diacylglycerol kinase | Inner membrane | 2.02 | 1.46 | — | — |
| Lipid transport and metabolism | fadI | 3-Ketoacyl-CoA thiolase | Cytoplasmic | 3.94 | 1.78 | 0.43 | 0.79 |
| | fadE | Acyl-CoA dehydrogenase | Inner membrane | 12.02 | 3.35 | 0.24 | 2.63 |
| | acs | Acetyl-CoA synthetase (AMP forming) | Cytoplasmic | 0.97 | 7.28 | 0.11 | 2.55 |
| | plsB | Glycerol-3-phosphate 1-O-acyltransferase | Inner membrane | — | — | 0.89 | 2.42 |
| Posttranslational modification, protein turnover, chaperones | gstA | Glutathione S-transferase | Cytoplasmic | — | — | 0.71 | 0.26 |
| | dsbG | Protein sulfenic acid reductase | Periplasmic | — | — | 0.27 | 0.14 |
| Posttranscriptional gene silencing by RNA | ohsC | Small regulatory RNA | Unknown | 0.35 | 0.73 | — | — |
| Replication, recombination, and repair | xthA | Exodeoxyribonuclease III | Cytoplasmic | 2.52 | 1.20 | 0.47 | 0.58 |
| | dusB | tRNA-dihydrouridine synthase B | Cytoplasmic | 4.35 | 1.80 | — | — |
| | rhlB | ATP-dependent RNA helicase | Cytoplasmic | — | — | 0.75 | 0.47 |
| | fimB | Regulator for fimA | Cytoplasmic | 0.12 | 0.15 | 0.09 | 0.13 |
| Nucleotide transport and metabolism | mtn | 5′-Methylthioadenosine/S-adenosylhomocysteine nucleosidase | Cytoplasmic | — | — | 0.91 | 0.40 |
| | dgt | dGTP triphosphohydrolase | Cytoplasmic | — | — | 0.77 | 0.29 |

**TABLE 2** (Continued)

| Gene category | Gene | Function[a] | Cellular location | Avg fold difference in expression | | | | |
|---|---|---|---|---|---|---|---|---|
| | | | | cpxA24 | ΔcpxA | pnlpE | ΔcpxR pnlpE | |
| Unknown function | yhdU | DUF2556 domain-containing protein | Inner membrane | 2.52 | 1.34 | 2.24 | 2.43 |
| | yncD | Putative TonB-dependent outer membrane receptor | Outer Membrane | 1.36 | 2.09 | — | — |
| | yncE | PQQ-like domain-containing protein | Unknown | 4.09 | 1.18 | 0.27 | 0.09 |
| | yfaH | Putative uncharacterized protein | Unknown | — | — | 1.08 | 7.71 |
| | yhdJ | DNA adenine methyltransferase | Cytoplasmic | — | — | 1.09 | 2.86 |
| | ygjR | Putative oxidoreductase YgjR | Unknown | — | — | 0.37 | 0.11 |
| | yibN | Putative sulfur transferase | Inner membrane | — | — | 0.55 | 0.20 |
| | yacH | DUF3300 domain-containing protein | Extracellular | — | — | 3.87 | 0.90 |

[a]CoA, coenzyme A; PTS, phosphotransferase system; PHB, poly-β-hydroxybutyrate.
[b]— means insignificant differences (< 2-fold) in the transcriptional levels.

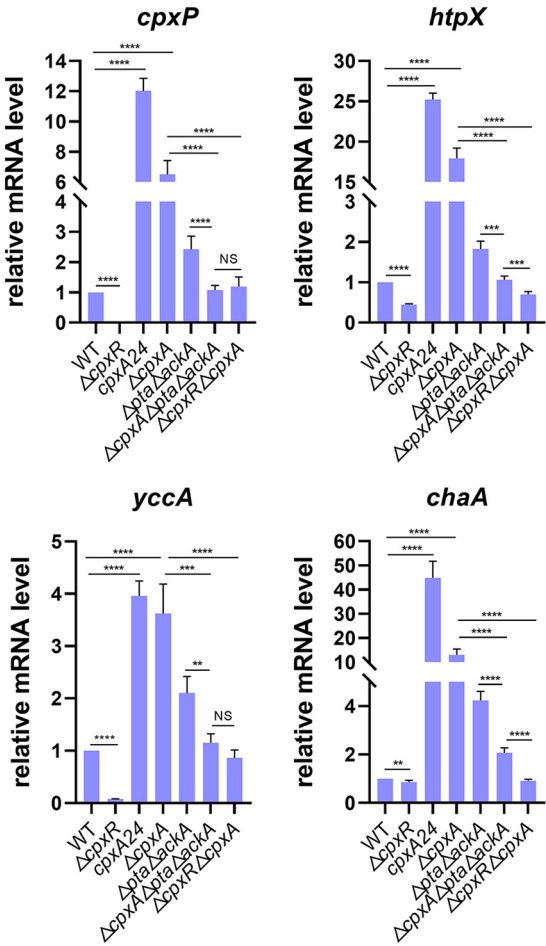

**FIG 2** The CpxR regulon is affected by the Pta-AckA pathway in BW25113. Relative mRNA levels of *cpxP*, *htpX*, *yccA*, and *chaA* were determined by qRT-PCR in the indicated strains. Statistical analysis was performed using a two-tailed Student's *t* test (**, $P < 0.01$; ***, $P < 0.001$; ****, $P < 0.0001$).

*chaA*, deletion of the Pta-AckA pathway in the Δ*cpxA* mutant still resulted in partial leaky output, implying that nonphosphorylated CpxR may also play a role in background induction. Alternatively, unidentified phosphoric acid contributors other than CpxA kinase and the Pta-AckA pathway are involved in CpxR protein activation. In addition, deleting the Pta-AckA pathway caused a slightly rise compared to the wild type, which could be reduced by further deleting *cpxA*, probably because the metabolic stress caused by the deletion of the Pta-AckA pathway would have a positive effect on CpxRA system activation.

Furthermore, we discovered significant differences (≥2-fold) in the transcriptional levels of 64 genes in another group of genetic backgrounds (Table 2). Similarly, these CpxR-regulated genes are involved in multiple physiological functions, and the majority of encoded proteins are located in the cytoplasm and inner membrane. Interestingly, in the strain carrying the p*nlpE* plasmid, even when the *cpxR* gene was deleted, some of these Cpx-regulated genes still showed an unusual response in the strain carrying the p*nlpE* plasmid, suggesting the existence of additional interactive pathways. For example, cross talk probably exists between different two-component signaling systems through phosphotransfer from a histidine kinase to a noncognate response regulator (29). Indeed, the histidine kinase CpxA has been shown to have cross-phosphorylation with OmpR, which is the response regulator of the EnvZ/OmpR TCS (30). As a result, when *nlpE* was overexpressed in the *cpxR* knockout mutant, transcription of some target genes was still higher/lower than in the wild-type strain, most likely due to OmpR phosphorylation.

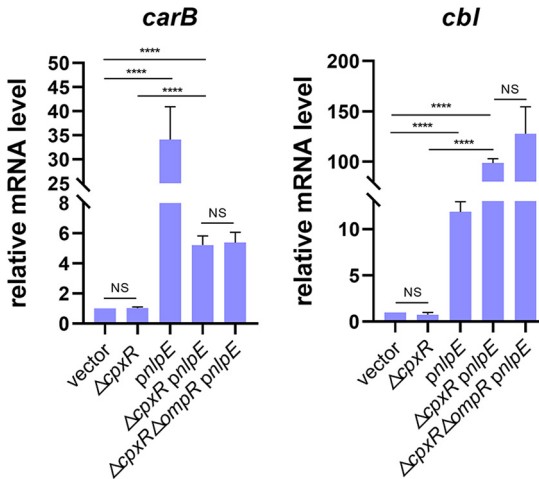

**FIG 3** The CpxR regulon is not affected by OmpR protein in BW25113. Relative mRNA levels of *craB* and *cbl* were determined by qRT-PCR in the indicated strains. Statistical analysis was performed using a two-tailed Student's *t* test (****, $P < 0.0001$; NS, no significance).

To test this hypothesis, we measured the mRNA levels of some CpxR-regulated genes in (i) BW25113 with the recombinant plasmid carrying *nlpE*, (ii) BW25113 with an empty vector, (iii) a Δ*cpxR* mutant with the recombinant plasmid carrying *nlpE*, or (iv) a Δ*cpxR* Δ*ompR* double mutant with a recombinant plasmid carrying *nlpE*. However, compared to the Δ*cpxR* mutant, the Δ*cpxR* Δ*ompR* double mutant did not result in significant changes in *carB* and *cbl* transcription upon NlpE overexpression (Fig. 3). Taken together, these results suggest that CpxA/OmpR cross talk is unlikely to play a role in regulating the expression of these genes.

If that is the case, what is the physiological reason for the higher/lower transcriptional level of the Δ*cpxR* p*nlpE* strain? Even though NlpE, as a sensor for multiple envelope stresses, has been exploited as a research tool to study Cpx in *E. coli* for a long time, the underlying signal transduction mechanism remained unclear. Delhaye et al. (31) demonstrated that NlpE specifically monitors lipoprotein sorting and oxidative folding as a sentinel and physically interacts with the CpxA through its N-terminal domain, while the interaction between NlpE and CpxA seems to be nonspecific. Overproduction of NlpE probably affects other signal transduction pathways besides CpxRA TCS and results in a complex effect on gene expression. For example, Feng et al. (32) recently proposed that the BaeSR two-component system is activated when NlpE detects a mechanical cue generated by initial host adherence. Given that NlpE overexpression displays complex effects besides activating the Cpx pathway, transcriptional analysis was performed in a Δ*cpxR* mutant with the empty vector. It was found the mRNAs of *carB* and *cbl* return to levels similar to that of the wild-type strain, with significant differences compared to the Δ*cpxR* p*nlpE* strain (Fig. 3). These findings suggest that NlpE overproduction affects gene expression in ways that are not entirely dependent on CpxR and that there is probably some cross talk and/or synergistic actions between the CpxRA system and other pathways after NlpE overproduction for the regulation of some specific target genes, which should be investigated further.

**CpxR binds to the promoter region of target genes in *E. coli* BW25113.** To explore whether CpxR regulates the expression of target genes identified in this study by directly binding to their promoter regions, we selected some candidates for testing by gel shift assay. For broad representation, 15 genes with diverse functions, corresponding to 11 potential CpxR-P recognition sites in various locations, were chosen at random from the four groups described above (Fig. 4A; Fig. S1A). The first group was represented by *carA-carB* and *dusB-fis*, where two predicted CpxR boxes exist within gene clusters (Fig. 4A). The second group was represented by *shoB-ohsC*, where two

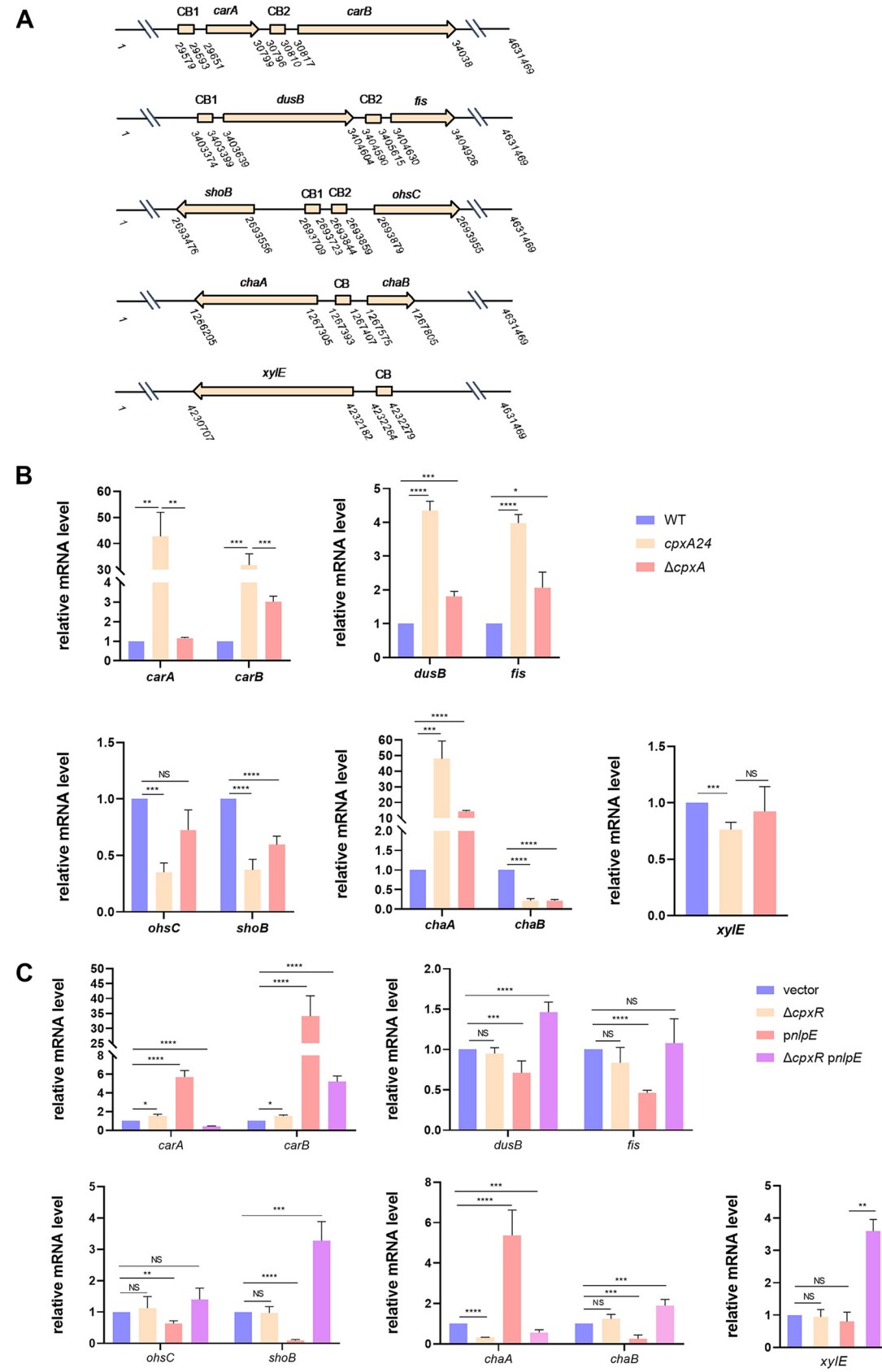

**FIG 4** Genomic locations and relative mRNA levels under two CpxRA activation conditions of some target genes. (A) Schematic diagram showing the genomic locations of target genes and corresponding CpxR boxes. (B and C) Relative mRNA

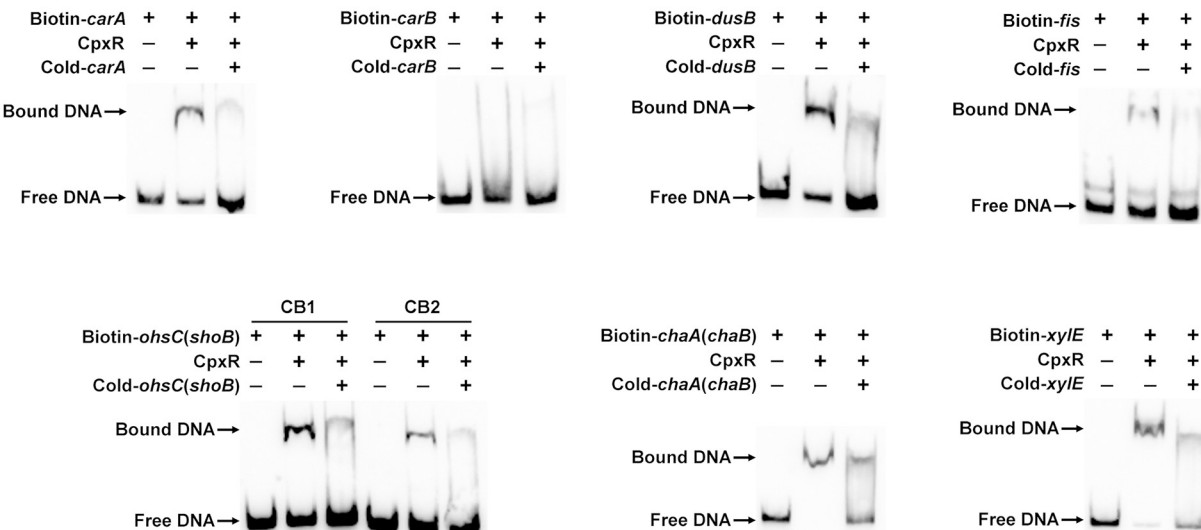

**FIG 5** CpxR regulates target genes through direct promoter binding. Gel shift assay, where biotin-labeled DNA fragments containing promoter regions of target genes were incubated without or with His$_6$-CpxR protein (lanes 1 and 2, respectively). Lane 3 is the same as lane 2 but supplemented with cold DNA fragments.

predicted CpxR boxes exist (Fig. 4A). The third group was represented by *chaA-chaB*, *fdnG-yddG*, *sbmA-ampH*, and *acs-nrfA*, with one predicted CpxR box in intragenic regions (Fig. 4A; Fig. S1A). The fourth group was represented by *xylE*, whose upstream region contained only one predicted CpxR box (Fig. 4A). qRT-PCR analysis showed that all of these genes were regulated by CpxR (≥2-fold) at least in one type of CpxRA activation condition (Fig. 4B and C; Fig. S1B and C). Also, purified His$_6$-CpxR protein could directly bind to DNA fragments corresponding to the promoter regions of these CpxR-regulated genes *in vitro*, and this was abolished by the addition of excess unlabeled competitor DNA (Fig. 5; Fig. S1D).

In the presence of multiple putative CpxR boxes, the gel shift analysis pointed to a CpxR-specific preference for one box over the other (Table S4; Fig. 5), which correlated with our PSSM scores. The positive correlation between the sequence score and the CpxR binding affinity suggests the usefulness of the PSSM method in predicting CpxR-regulated operons. Particularly, for the first group of genes like *carA-carB* and *dusB-fis*, CpxR may be more likely to bind to the recognition site in front of the gene cluster (CB1) and regulate the expression of genes (Fig. 4A). However, despite the observed correlation between CpxR binding and PSSM scores, the latter are not a complete indicator of the CpxR regulatory ability for candidate genes. For example, upstream of the *proP* and *adiA* genes, sequences with high scores of 16.75 and 14.82, respectively (Table S4), were found, and EMSA analysis revealed that CpxR directly binds to *proP* and *adiA* promoters (Fig. S2). However, qRT-PCR analysis showed no significant difference (<2-fold) in the expressions of *proP* and *adiA* under both Cpx pathway-activating conditions (data not shown). This could be because CpxR binding to the site upstream of *proP* and *adiA* may require additional activation conditions to result in *in vivo* regulation. Thus, although a rough prediction of target operons in *E. coli* appears to be reasonable using PSSM, and more than half of these sites (73/97) are functional, questionable candidates must be identified and subjected to additional testing, such as transcriptional analysis or *in vitro* DNA binding analysis, for more conclusive identification of CpxR-regulated genes under different activating conditions of CpxRA.

**FIG 4** Legend (Continued)

levels of target genes obtained by qRT-PCR in BW25113 and the *cpxA24* and Δ*cpxA* mutants (B) and in strains carrying empty vector or p*nlpE* and the Δ*cpxR* mutant carrying p*nlpE* (C). Statistical analysis was performed using a two-tailed Student's *t* test (*, $P < 0.05$; **, $P < 0.01$; ***, $P < 0.001$; ****, $P < 0.0001$; NS, no significance).

**TABLE 3** Candidate genes in acid tolerance or protamine resistance

| Category and gene | Description | Reference(s) |
|---|---|---|
| Acid tolerance | | |
| ydeE | Member of the DHA family within the MFS of transporters involved in dipeptide and arabinose export and dipeptide resistance | 42, 59, 60 |
| frc | Required during the adaption phase of an oxalate-induced acid tolerance response | 61 |
| xylE | D-Xylose/proton symporter which can elicit an alkaline pH change; a member of the MFS of transporters | 34, 35 |
| chaA | Na$^+$/K$^+$:proton antiporter implicated in proton uptake at alkaline pH >8 | 62 |
| bluR | Repressor for acid resistance genes | 63 |
| focA | Functions as a channel and may undergo pH-dependent gating | 64, 65 |
| gudP | Potential D-glucarate or galactarate transporter; a member of the MFS of transporters | 42, 66 |
| alx | Expression is repressed by low pH under aerobic conditions | 41 |
| galP | Galactose:H$^+$ symporter; member of the MFS of transporters | 37 |
| Protamine resistance | | |
| prlF | Antitoxin component in the PrlF-YhaV antitoxin-toxin complex | 67 |
| ydeE | Assumed to be a drug transporter on the basis of sequence similarities | 68 |
| casA | Regulated by BaeSR, which increases the novobiocin and deoxycholate resistance of *E. coli* | 69 |
| yacH | Transcription is reduced upon exposure to a sublethal dose of the cationic antimicrobial insect peptide cecropin A | 70 |
| sbmA | Transports a peptide antibiotic | 44 |
| ampH | Penicillin-binding protein that catalyzes both DD-carboxypeptidase and DD-endopeptidase activities | 46, 47 |

**CpxR-regulated genes contribute to *E. coli* resistance to acid stress.** The CpxRA system could be activated by mild acid stress and activates transcription of *fabA* and *fabB* genes, which are essential in the biosynthesis of unsaturated fatty acids (UFAs). Increased UFA production improves bacterial tolerance to acid stress (7). To further identify more CpxR-regulated genes contributing to *E. coli* resistance to acid stress, we measured the survival of several single-deletion strains during exponential growth after an acid challenge. In either *cpxA24* or NlpE overexpression backgrounds, these target genes showed high or moderate regulation and were most likely involved in acid tolerance, according to their function (Table 3). The exponentially growing *E. coli* BW25113 wild-type strain and single-deletion strains (from the Keio collection [33]) were transferred into minimal medium E at pH 7.0 or pH 3.0. As in our previous study (7), the survival was calculated by determining numbers of CFU of *E. coli* growing at pH 3 versus CFU of *E. coli* growing at pH 7.0, which represents the acid tolerance of exponentially growing *E. coli*. Under our growth conditions, acid stress produced clear effects on the survival of *cpxR*, *ydeE*, *xylE*, *alx*, and *galP*, mutant strains compared to the wild type ($P < 0.05$) (Fig. 6A). Specifically, deletion of the *cpxR* gene reduced the CFU ratio after acid challenge, confirming that CpxRA is required for the response to the acidic challenge. Also, *ydeE* and *xylE* are likely protective against acid resistance, whereas *alx* and *galP* had the opposite effect. The discovery of these acid resistance genes provides new insights into the acid tolerance mechanism in *E. coli*.

Given that CpxRA regulates target genes depending on different environmental stimulus, we measured the relative transcription level of the acid-related genes identified in our study. The mRNA level of *cpxP*, which encodes a small periplasmic protein, was induced in a Cpx-dependent manner and increased significantly after acidic challenge (Fig. 6B), indicating activation of the CpxRA TCS. qRT-PCR analysis showed that the acidic stress could activate *xylE* transcription and inhibit *galP* transcription (Fig. 6B), whereas deleting *cpxR* partially alleviated the effects of acid challenge on gene expression. These results indicate that their positive and negative effects on acid resistance are both dependent on the CpxRA system and that other unidentified pathways must regulate *xylE* and *galP* expression after acidic stress. XylE is a D-xylose/proton symporter which can elicit an alkaline pH change (34) and is a member of the major facilitator superfamily (MFS) of transporters (35).

Uphill transport appears to be energized by a proton-motive force (36). Similarly, GalP is a galactose:H$^+$ symporter and also belongs to the MFS (37). This protein has

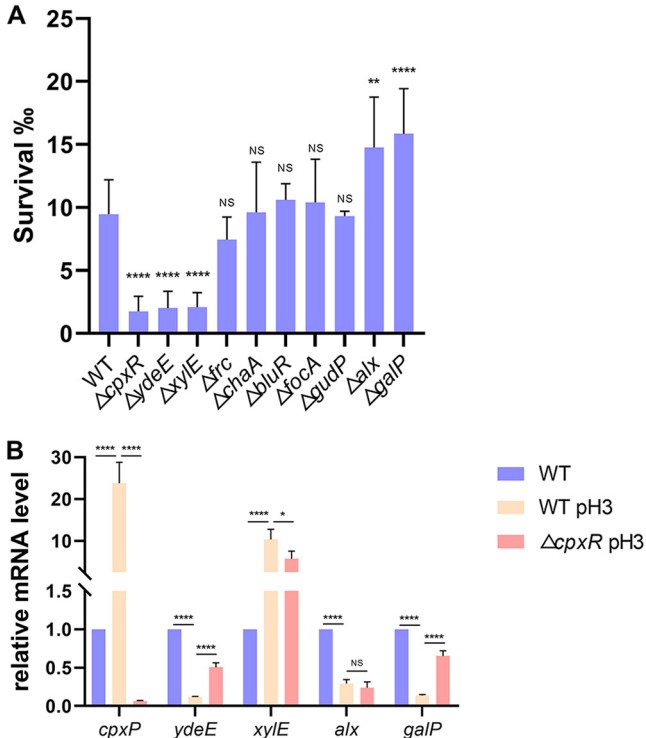

**FIG 6** CpxR-regulated genes contribute to acid resistance. (A) Growth of the *E. coli* BW25113 wild-type strain and single-deletion strains after acidic challenge at pH 3. Strain BW25113 was used as the control. (B) Relative mRNA levels of target genes determined by qRT-PCR in the *E. coli* BW25113 wild-type strain and Δ*cpxR* strain after acidic challenge at pH 3. Statistical analysis was performed using a two-tailed Student's *t* test (*, $P < 0.05$; **, $P < 0.01$; ****, $P < 0.0001$; NS, no significance).

been shown to share a high level of sequence similarity with XylE (34% identity) in *E. coli* (38). Although XylE and GalP are both involved in proton transport, it is unclear whether they can transport $H^+$ in the absence of xylose or galactose. Together, our results suggest that both genes are likely functionally related to the acid tolerance response in an CpxRA-dependent manner. As *alx* is a known CpxR-regulated gene (21), its expression can respond to changes in pH, and it could be highly induced by alkaline pH (8.5 and above), both aerobically and anaerobically (39, 40), and repressed by acidic pH only aerobically (41). However, whether CpxRA plays a role in acid resistance by downregulating *alx* expression is not clear. Although deletion of *alx* increased the CFU ratio compared to the wild type (Fig. 6A) and acidic stress inhibited *alx* transcription as previously reported (Fig. 6B), deleting *cpxR* did not increase the expression of *alx* (Fig. 6B). This indicates that *alx* does contribute to acid resistance in *E. coli*, but the effects of other unknown regulatory systems probably act on *alx* expression and may mask the effect of the CpxRA system after an acidic challenge. In addition, *ydeE* encodes a protein which is a putative member of the drug:$H^+$ antiporter-1 (DHA) family (42). The Δ*ydeE* mutant appears to have a lower survival rate than the wild type (Fig. 6A), but the acid challenge resulted in a downregulation of *ydeE* gene transcription, and this inhibition was partially alleviated by deleting *cpxR*.

These results suggested that *ydeE* contributes to *E. coli* resistance to acid stress and that the acid response of *ydeE* gene is dependent on the CpxRA system; however, the acid response mechanism of *ydeE* and whether other signal transduction pathways affect *ydeE* expression are both unknown. Thus, although our study is the first to identify four CpxR-regulated genes related to the *E. coli* acidic stress response, the expression of target genes after an acid challenge is likely to be a complex, multifactorial trait, because numerous cellular functions are impacted by the Cpx pathway. Furthermore, additional signal transduction pathways involved in the acid stress response probably

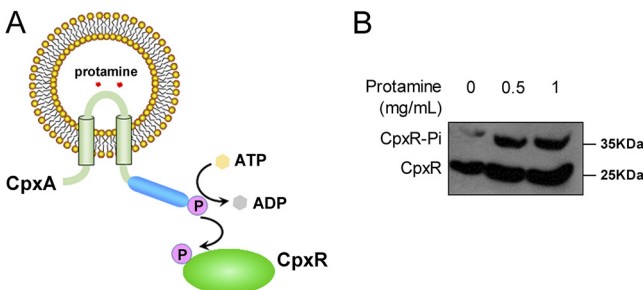

**FIG 7** The CpxRA system is activated by protamine stress. (A) Schematic diagram of the proteoliposome system. (B) Reconstituted proteoliposomes analysis of CpxR phosphorylation induced by protamine *in vitro*. Reconstituted proteoliposomes were preloaded with buffer at different concentrations (0, 0.5, and 1.0 mg/mL) of protamine. Purified His$_6$-CpxR was incubated with CpxA-His$_6$-containing proteoliposomes in phosphorylation buffer.

interact with CpxRA in a complex cross talk or directly affect the expression of target genes as transcriptional regulators to protect exponentially growing *E. coli* from acidic stress.

**CpxR-regulated genes contribute to *E. coli* resistance to protamine.** A previous study proposed that the CpxRA system facilitates bacterial resistance to protamine, a model cationic antimicrobial peptide (CAMP) (43). To understand whether the CpxRA TCS senses external protamine directly, we examined the phosphorylation level of CpxR at different concentrations of protamine (0, 0.5, and 1.0 mg/mL) using an *in vitro* reconstituted proteoliposome system (7) (Fig. 7A). In this study, purified CpxA-His$_6$ protein was reconstituted into vesicles in the inside-out orientation. The increased CpxR-P levels were accompanied by a higher protamine concentration inside vesicles (Fig. 7B), suggesting that protamine stress directly activates the periplasmic domain of CpxA kinase *in vitro*, similarly to the CpxA response to acid stress (7).

To explore new CpxR-regulated genes that contribute to *E. coli* resistance to protamine, we carried out a susceptibility assay to compare the protamine resistance of *E. coli* BW25113 wild-type and single-deletion strains from the Keio collection (33). All of these target genes were CpxR-regulated genes and were likely involved in protamine tolerance, according to their function (Table 3). The results showed that protamine killed the *E. coli* wild-type strain in a concentration-dependent manner (Fig. 8A). Furthermore, compared to the *E. coli* wild-type strain, the Δ*cpxR* mutant was more susceptible to high concentration protamine, indicating the role of CpxRA system in bacterial resistance to protamine (Fig. 8B). However, at low protamine concentrations, no significant differences were observed between the Δ*cpxR* mutant and wild-type strain. Surprisingly, other single-deletion strains (Δ*sbmA*, Δ*prlF*, Δ*casA*, Δ*ampH*, Δ*yacH*, and Δ*ydeE* mutants) outperformed the wild-type strain in terms of survival after protamine challenge (Fig. 8A), suggesting that these target genes contribute to *E. coli* resistance to protamine.

Next, we attempted to investigate the roles of these CpxR-regulated genes in *E. coli* protamine resistance. qRT-PCR results showed that protamine significantly increased the expression of *cpxP* of the wild-type strain (Fig. 8B) in a concentration-dependent manner (Fig. 8C), suggesting that the CpxRA system is activated by protamine stress. In contrast, the mRNA levels of *casA*, *prlF*, *yacH*, and *ydeE* were reduced in a protamine concentration-dependent manner compared to untreated control bacteria (Fig. 8B and C). However, when the *cpxR* gene was deleted, this transcriptional depression caused by protamine stress was partially alleviated (Fig. 8B). These findings demonstrated that *prlF*, *casA*, *yacH*, and *ydeE* are likely involved in *E. coli* protamine resistance, which is dependent on the CpxRA system acting as an inhibitor. Surprisingly, the protamine stress inhibits the transcription of *casA* and *yacH* genes (Fig. 8B and C), which were both activated under conditions of *cpxA24* mutation or NlpE overexpression (Table 2). The probable reason is that the expression pattern of the CpxR-regulated genes under different

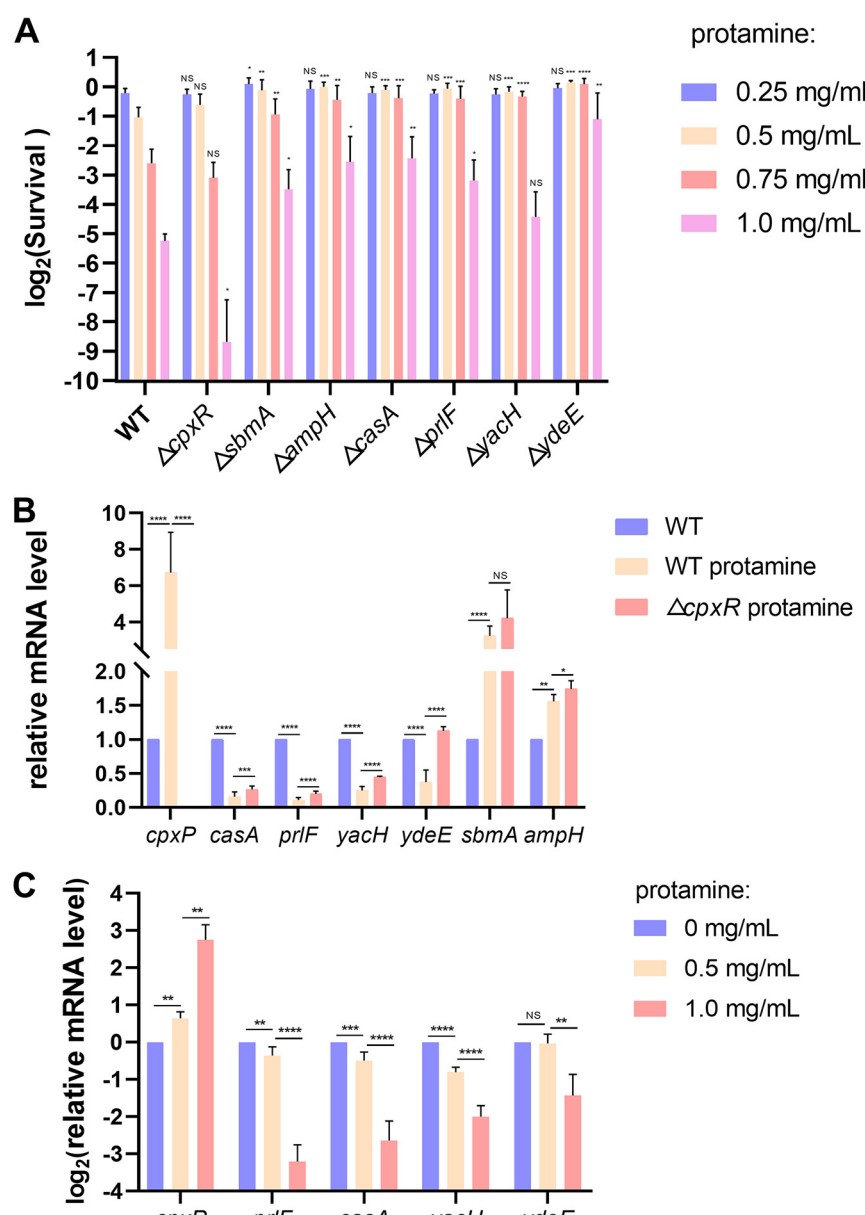

**FIG 8** CpxR-regulated genes contribute to protamine resistance. (A) Protamine susceptibility assay for the wild-type strain and Δ*cpxR*, Δ*sbmA*, Δ*prlF*, Δ*casA*, Δ*ampH*, Δ*yacH*, and Δ*ydeE* single mutants on LB plates containing protamine (0.25, 0.5, 0.75, or 1.0 mg/mL). The corresponding concentration of strain BW25113 was used as the control. (B) Relative mRNA levels of target genes in the *E. coli* BW25113 wild-type strain and Δ*cpxR* strain after stimulation with or without 1.0 mg/mL protamine. (C) Relative mRNA levels of target genes in the wild-type strain after stimulation with the indicated protamine concentrations. Statistical analysis was performed using a two-tailed Student's *t* test (*, *P* < 0.05; **, *P* < 0.01; ***, *P* < 0.001; ****, *P* < 0.0001; NS, no significance).

activation conditions is not always constant. It is most likely a fine regulation dependent on the CpxRA system in response to various signal stimulations. Concurrently, unknown pathways may cross-regulate these target genes in conjunction with the Cpx pathway, providing synergistic defense against protamine challenge. Taken together, our results suggest that *prlF*, *casA*, *yacH*, and *ydeE* all play roles in protamine resistance in a partially CpxRA-dependent manner. which will provide new insights into the mechanism of CAMP resistance in *E. coli* and the cross talk between various signal transduction pathways.

Interestingly, in these CpxR-regulated genes, *ydeE* is a linking target gene that contributes to both protamine and acid resistance. Furthermore, although Δ*sbmA* and

Δ*ampH* strains showed higher resistance to protamine (Fig. 8A), the transcription levels of these genes increased after the protamine challenge, regardless of the presence of the CpxRA system (Fig. 8B). *sbmA* is a known CpxR-regulated gene (15) that encodes an inner-membrane transport protein that is responsible for the import of microcin 25, an antibiotic peptide (44), and plays a significant role in antibiotic bleomycin resistance (45), whereas *ampH* encodes a penicillin-binding protein that is probably involved in peptidoglycan remodeling and/or recycling (46, 47). We demonstrated that they were both involved in protamine resistance in *E. coli*, but whether this physiological process is linked to the Cpx pathway is still unclear.

Due to the interesting correlation between CpxRA-mediated protamine resistance and acid resistance, other linking CpxR-regulated genes besides *ydeE* may also contribute to these two environmental stress responses. Thus, we attempted to determine whether other acid resistance-related genes identified in this study (*xylE*, *alx*, and *galP*) are involved in the protamine stress response. The Δ*alx* mutant displayed decreased susceptibility to protamine (Fig. S3A). qRT-PCR analysis showed that protamine stress can activate the transcription of *alx*, where the CpxRA system acts as an activator (Fig. S3B). However, the regulatory pathway of *alx* in response to protamine stress is not clear. Overall, these results suggest a potential link between these two environmental stress responses.

## DISCUSSION

The CpxRA system is a well-known TCS that responds to several environment-associated simulations and protects cells against a wide variety of surrounding stresses. As a typical TCS, the process of CpxA autophosphorylation at a specific histidine and phosphoryl group transfer to an aspartate residue of CpxR has been well elaborated (48). However, the identification of CpxR-regulated genes and the analysis of their function in response to environmental stress is grossly inadequate. In fact, a critical factor in understanding how bacteria employ the CpxRA TCS against environmental stress lies in the identification of CpxR-regulated genes.

Our study used PSSM, a bioinformatics analysis for sequence-based prediction, to systematically screen genome-wide profiling of CpxR promoters. Using alignments with data for 41 known CpxR binding sequences, thousands of putative CpxR binding sites (6,522 conserved 15-bp sequences and 6,464 conserved 16-bp sequences) were obtained. This bioinformatics analysis reveals a referential view of targets based on sequence and provides a molecular basis for identification of CpxR-regulated genes. The potential binding sites for CpxR are distributed evenly across the *E. coli* BW25113 chromosome (Fig. 9) The distribution of the 97 candidate genes we selected randomly in this study is not skewed across the genome (Fig. 9), and we found that 73 of these were controlled by the Cpx pathway. This greatly increases the number of known CpxR-regulated genes and potentially enables the discovery of the complex interrelationships between the CpxRA system and other regulatory pathways.

Each putative CpxR binding site with a high PSSM score ($>$17.68) was found to correspond to at least one CpxR-regulated gene (Table 2; Table S4). Further, when more than one CpxR binding site was predicted in the promoter regions of candidate genes, gel shift assays showed preferential CpxR binding to putative CpxR boxes with higher PSSM scores. These results support the use of PSSM scores as a tool in the identification of target genes, based on the recognition of specific binding sequences for a regulator protein. Additionally, the discovery of 73 new CpxR-regulated genes enhances sequence-based data sets. Evolutionary information can further improve the prediction capacity of the PSSM method, allowing circular screening of the genome sequence and the identification of more meaningful CpxR binding sites.

As reported previously, the CpxRA is a key system in the acid and CAMP stress responses of exponentially growing *E. coli* (6, 7). Herein, several CpxR-regulated genes were shown to be involved in *E. coli* exponential-phase survival after challenges with acid or CAMP. Although their regulatory mechanisms and their effects on cellular

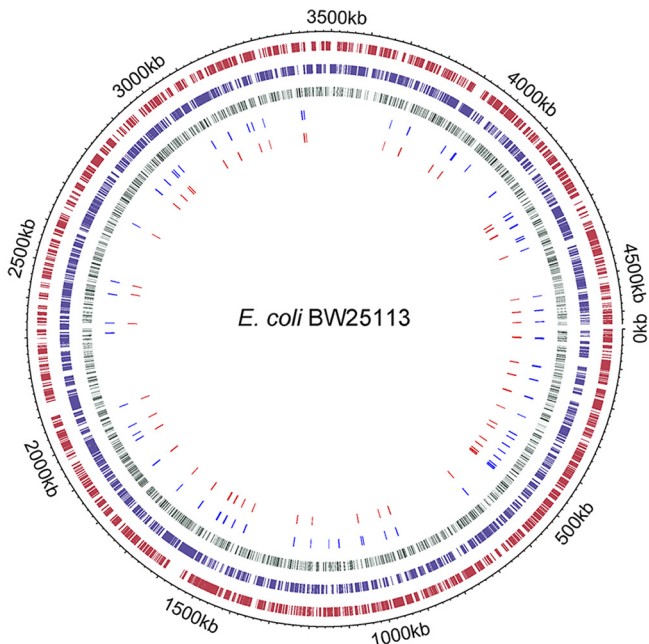

**FIG 9** Visualization of the whole genome by using Circos. From outermost to innermost, the layers represent chromosomes of *E. coli* BW25113, the locations of the genes in *E. coli* BW25113 (dark red), the location of complementary chain's genes in *E. coli* BW25113 (dark blue), and the location of putative CpxR binding sites across the *E. coli* BW25113 chromosome, with the depth of color being proportional to the PSSM scores of sequences (green), 91 candidate CpxR-regulated genes we selected in this study (blue), and 67 CpxR-regulated genes we identified in this study (red). Circos analysis was performed using the OmicStudio tools at https://www.omicstudio.cn/tool/.

physiological functions remain unknown, the discovery of nine target genes involved in these two stress responses will provide new insights into CpxRA-dependent resistance mechanisms. For the first time, the linkages between these two stress resistances are proposed in this study, suggesting that synergistic effects probably exist in a CpxRA-mediated multiple signal transduction pathway. Furthermore, Raivio has proposed that the CpxRA system appears to play a role in altering inner membrane transport in all cases studied thus far (49). Thus, in future studies, to gain a better understanding of *E. coli* resistance mechanisms to acid and antimicrobial peptides, we hope to focus on more CpxR-regulated genes that are related to proton or toxic-compound transport.

Surprisingly, the environment or genetic background appears to have a strong influence on the expression of CpxR-regulated genes. Indeed, constitutively activated *cpxA* mutation and NlpE overexpression exposed different CpxR-regulated genes, suggesting that neither approach was exhaustive. Also, after activation of the CpxRA system, the expression pattern of these regulated genes is not constant. As a result, the expression of CpxR-regulated genes varies depending on the activation condition (*cpxA* mutation, acidic conditions, CAMP, or NlpE overexpression). For example, NlpE overproduction inhibited *xylE* expression, but acidic challenge increased its transcriptional level (Table 2; Fig. 6B). Also, the expression of *casA* was activated in *cpxA24* mutant or after NlpE overexpression but was inhibited under a protamine challenge (Table 2; Fig. 8B and C). In addition, Raivio noted that the Cpx response and other cellular signaling pathways may have complex connections, such as feed-forward and feedback inhibition loops, which may enhance the precision and/or magnitude with which the Cpx response affects adaptive gene expression (49). As a result, the stress response of some genes may be the result of synergistic fine regulation of a combination of regulatory pathways. Our results indicated that the regulatory mechanism of the CpxRA TCS is complex and dynamic and is dependent on environmental cues and the genetic

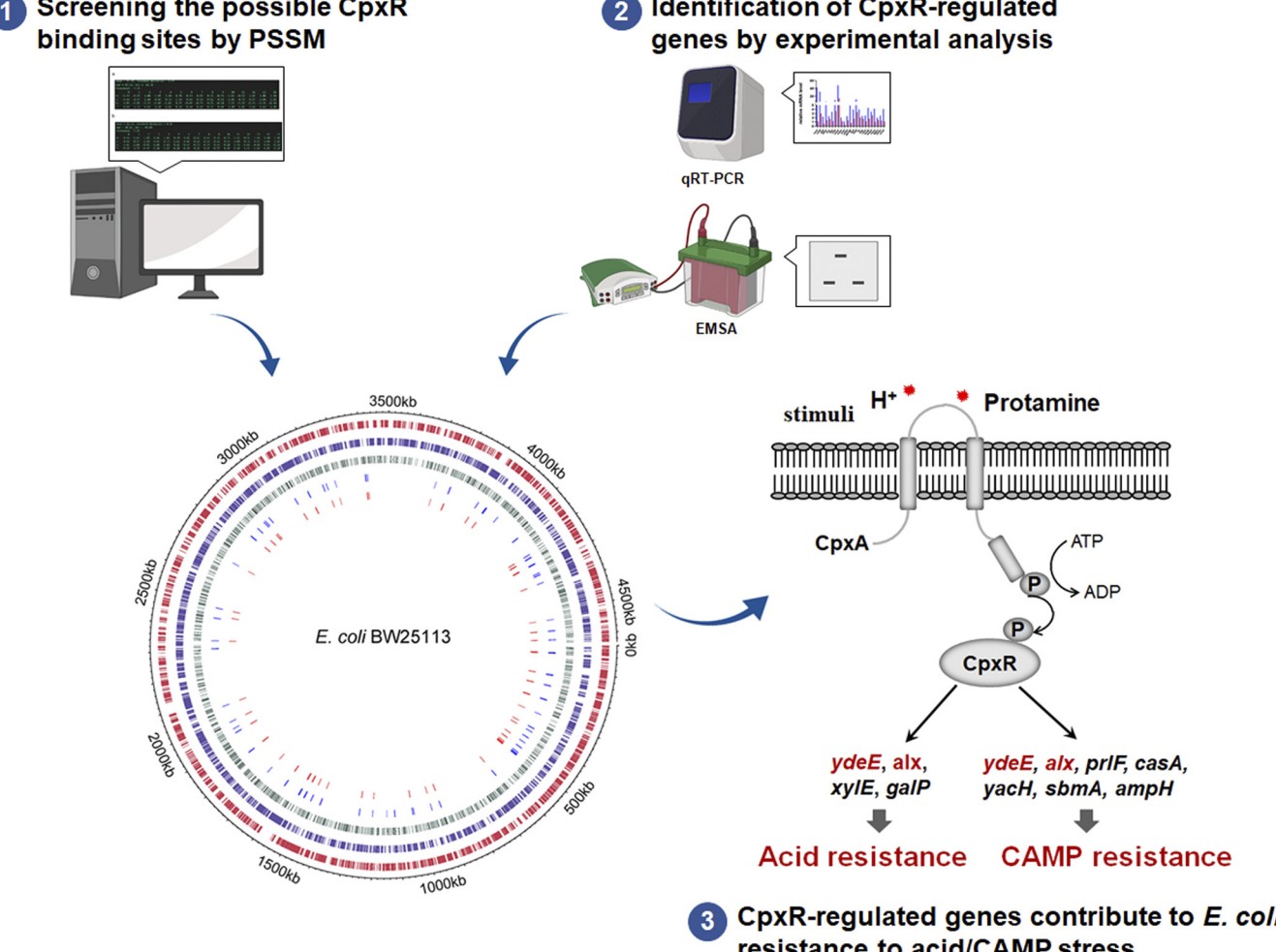

**FIG 10** Systematic identification of CpxRA-regulated genes based on bioinformatics technology and experimental analysis and their roles in *E. coli* stress response.

background of bacteria. This may be very important for environment-bacterium interaction and bacterial evolution in a specific environment and could also guide future studies aimed at uncovering the cross talk of different signaling pathways or regulator factors.

In summary, PSSM predicted a large number of putative CpxR binding sites based on sequence characteristics, and the matrix score correlated with the relative site affinity for CpxR protein *in vitro*. A series of new CpxR-regulated genes were determined under two activation conditions of the Cpx pathway. The acid and CAMP resistance-related genes controlled by the CpxRA system can help elucidate the mechanism of environmental stress responses (Fig. 10). Remarkably, the availability and efficiency of PSSM based on genome screening have facilitated the discovery of novel putative CpxR recognition sites, which in combination with experimental analysis will vastly improve our understanding of how bacteria respond to environmental signals by regulating various genes, and this may serve as a typical method to develop sequence-specific screens for the novel regulon's identification of various TCSs in different bacteria. Taken together, these results provide more insights into the *E. coli* stress response network dependent on the CpxRA system and offer us potential targets that can be used for combating infection.

## MATERIALS AND METHODS

**Bacterial strains and growth conditions.** All strains used in this study are listed in Table S5, and all primers used are listed in Table S6. Bacteria were grown at 37°C in Luria-Bertani broth or in E minimal

medium (0.8 mM $MgSO_4$, 10 mM citric acid, 57.5 mM $K_2HPO_4$, 16.7 mM $NaNH_4HPO_4$, 0.5% glucose). When necessary, antibiotics were added at final concentrations of 100 $\mu$g/mL for ampicillin. *E. coli* DH5$\alpha$ was used as a host for the preparation of plasmid DNA, and *E. coli* $\chi$7213 was used for the preparation of suicide vectors. Diaminopimelic acid (DAP) (50 $\mu$g/mL) was used for the growth of $\chi$7213 strain. LB agar containing 10% sucrose was used for *sacB* gene-based counterselection in allelic exchange experiments.

In this study, *E. coli* BW25113 *cpxA24* was constructed by homologous recombination using a suicide plasmid. *E. coli* BW25113$\Delta$*cpxR*$\Delta$*cpxA*, *E. coli* BW25113$\Delta$*pta*$\Delta$*ackA*, *E. coli* BW25113$\Delta$*cpxA*$\Delta$*pta*$\Delta$*ackA*, and *E. coli* BW25113$\Delta$*cpxR*$\Delta$*ompR* were constructed using the $\lambda$ Red recombinase system (50).

In the experiment on activation of the Cpx pathway by the *cpxA24* mutant, the strains were grown at 37°C in Luria-Bertani broth to exponential phase. In the experiment on activation of the Cpx pathway by overexpression of NlpE, the strains were grown in E medium (pH 7.0), IPTG (isopropyl-$\beta$-D-thiogalactopyranoside) was added to a final concentration of 0.5 mM at an optical density at 600 nm ($OD_{600}$) of 0.4, and the strain was further grown to an $OD_{600}$ of 0.6.

**Position-specific scoring matrix screening.** PSSMs were calculated with the Python tool package Biopython (51), assuming that the probabilities for each position are statistically independent with a pseudocount of 0.5. The matrix screening method predicted the affinity of CpxR for DNA sequences in the genome of *E. coli* BW25113 (GenBank accession no. CP009273) based on the sequence statistics of 41 known CpxR binding sites. The score for all continuous 15-bp sequences in the genome was calculated, and the scores higher than the cutoff were considered potential CpxR binding sites. For 16-bp sequences, the middle base of $N_5$ was duplicated to yield the data set for $N_6$.

**Quantitative RT-PCR.** Total RNA was isolated from bacterial culture using an EASYSpin Plus bacterial RNA quick extraction kit (Aidlab Biotechnologies, China) according to the manufacturer's instructions. RNA concentration was determined by spectrophotometry at 260 nm. Removal of genomic DNA and synthesis of cDNA were carried out using a PrimeScript RT reagent kit with gDNA Eraser (TaKaRa, Japan). qRT-PCR was conducted using TB Green Premix Ex Taq (TaKaRa, Japan) with the QuantStudio 1 system (Applied Biosystems, USA). The constitutively transcribed gene *rpoD* was used as a reference control to normalize the total RNA quantity of different samples. Differences between mRNA levels were calculated using the $\Delta\Delta C_T$ method (52). Two independent biological samples with three technical repeats for each sample were performed for each qRT-PCR analysis.

**EMSA.** Purification of $His_6$-CpxR was conducted according to our previous work (7). Primers were labeled using biotin. The promoter regions of target genes were amplified with primers listed in Table S6. Biotin-labeled DNA (0.1 pM) was incubated at room temperature for 30 min with 0 or 60 pmol of $His_6$-CpxR protein in 20 $\mu$L of an EMSA buffer (Beyotime, China). The mixture was subjected directly to 6.5% Tris-borate-EDTA (TBE)-PAGE. Signals were detected with a luminometer.

**pH sensitivity assay.** Specific strains from the Keio collection (33) were selected to test their susceptibility to an acid condition. Bacterial cells were cultured overnight, harvested, and washed twice with double-distilled water ($ddH_2O$), reinoculated (1:100) in E medium (pH 7.0), and grown to an $OD_{600}$ of 0.6. Cells were harvested and washed twice with $ddH_2O$, and inoculated into E medium at various pHs, as indicated, and strains were grown for another 1 to 2 h before the cells were collected to determine the number of CFU.

**Protamine susceptibility assay.** Specific strains from the Keio collection (33) were selected in order to test their susceptibility to protamine. Bacterial cells were cultured overnight, reinoculated (1:100) in LB broth, and grown for 3 h at 37°C. Cultures were diluted, inoculated dropwise onto LB agar plates containing various concentrations (0.25 to 1.0 mg/mL) of protamine sulfate (Aladdin, China), and incubated overnight at 37°C to determine the number CFU. The percentage survival of each strain was calculated by comparing numbers of CFU from plates supplemented with and without protamine.

**Preparation of proteoliposomes.** Purification of His6-CpxR and CpxA-His6 and reconstitution in proteoliposomes were performed as previously described (7). Briefly, *E. coli* phospholipids (Avanti, USA) were dried under a stream of nitrogen gas and slowly dissolved in sodium citrate-hydrochloric acid buffer (pH 7.0) with 10% glycerol (vol/vol), 0.47% Triton X-100 (vol/vol), and various concentrations (0, 0.5, and 1.0 mg/mL) of protamine. Purified CpxA-$His_6$ was added to the mixture at a phospholipid/protein ratio of 100:1 (wt/wt) and stirred at room temperature for 20 min. Bio-Beads SM-2 (Bio-Rad) were added at a bead/detergent ratio of 10:1 (wt/wt), and the mixture was gently stirred at 4°C overnight. After 16 h, fresh Bio-Beads were added, and the mixture was stirred for another 6 h. Proteoliposomes were collected by ultracentrifugation and then incubated with 300 $\mu$mol ATP in phosphorylation buffer (50 mM Tris-HCl [pH 7.5], 10% glycerol [vol/vol], 2 mM dithiothreitol, 50 mM KCl, 5 mM $MgCl_2$) at room temperature for 30 min. A 5$\times$ SDS sample buffer was loaded to terminate the reaction. Purified $His_6$-CpxR was added to this mixture. The samples were ultracentrifuged after 20 min reaction, and the upper phase was collected. The 5$\times$ SDS sample buffer was added to stop the reaction. To detect the phosphorylation level of CpxR, all the samples were subjected to 8% SDS-PAGE with 20 to 50 $\mu$M Phos-tag acrylamide (Wako) and 0.1 mM $Mn^{2+}$.

## SUPPLEMENTAL MATERIAL

Supplemental material is available online only.

**FIG S1**, TIF file, 0.8 MB.
**FIG S2**, TIF file, 0.1 MB.
**FIG S3**, TIF file, 0.3 MB.

**TABLE S1**, DOCX file, 0.04 MB.
**TABLE S2**, CSV file, 0.3 MB.
**TABLE S3**, CSV file, 0.4 MB.
**TABLE S4**, DOCX file, 0.04 MB.
**TABLE S5**, DOCX file, 0.02 MB.
**TABLE S6**, DOCX file, 0.04 MB.

## ACKNOWLEDGMENTS

This work was supported by the National Key Research and Development Program of China (2021YFC2100503), National Natural Science Foundation of China (32170085 and 31961133014), and Foundation for Innovative Research Groups of State Key Laboratory of Microbial Technology, Distinguished Young Scholars Program of Shandong University (G.Z.).

G.Z. designed the experiments. Z.Z., Y.X., B.J., and J.W. performed the experiments. G.Z., M.X., Q.Q., Y.-J.T., Z.Z. and J.W. analyzed the results. G.Z., Z.Z. and J.W. wrote the manuscript. All authors edited the manuscript before submission.

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
