## [Reviewer comments · mSystems]

Systematic identification of CpxRA-regulated genes and their roles in *Escherichia coli* stress response

Guang Zhao, Zhe Zhao, Ying Xu, Qingsheng Qi, Ya-Jie Tang, Mo Xian, Jichao Wang, and Bo Jiang

Corresponding Author(s): Guang Zhao, Shandong University

Review Timeline:

Submission Date:

May 24, 2022

Accepted:

July 10, 2022

Editor: Youjun Feng

Reviewer(s): Disclosure of reviewer identity is with reference to reviewer comments included in decision letter(s). The following individuals involved in review of your submission have agreed to reveal their identity: Bin Liu (Reviewer #1)

Transaction Report:

DOI: <https://doi.org/10.1128/msystems.00419-22>

July 10, 2022

Prof. Guang Zhao
Shandong University
72 Binhai Road
Qingdao, Shandong 266237
China

Re: mSystems00419-22 (Systematic identification of CpxRA-regulated genes and their roles in *Escherichia coli* stress response)

Dear Prof. Guang Zhao:

Your manuscript has been accepted, and I am forwarding it to the ASM Journals Department for publication. For your reference, ASM Journals' address is given below. Before it can be scheduled for publication, your manuscript will be checked by the mSystems production staff to make sure that all elements meet the technical requirements for publication. They will contact you if anything needs to be revised before copyediting and production can begin. Otherwise, you will be notified when your proofs are ready to be viewed.

Publication Fees:

If you would like to submit a potential Featured Image, please email a file and a short legend to mssystems@asmusa.org. Please note that we can only consider images that (i) the authors created or own and (ii) have not been previously published. By submitting, you agree that the image can be used under the same terms as the published article. File requirements: square dimensions (4" x 4"), 300 dpi resolution, RGB colorspace, TIF file format.

We recognize that the video files can become quite large, and so to avoid quality loss ASM suggests sending the video file via <https://www.wetransfer.com/>. When you have a final version of the video and the still ready to share, please send it to mSystems staff at mssystems@asmusa.org.

Sincerely,

Youjun Feng
Editor, mSystems

Journals Department
Fig. S1: Accept
Table S2: Accept
Table S1: Accept
Table S5: Accept
Table S4: Accept
Fig. S3: Accept
Table S6: Accept
Table S3: Accept
Fig. S2: Accept